# Evidence of health system resilience in Myanmar during Cyclone Nargis: a qualitative analysis

Pauline Yongeun Grimm [ORCID],[1,2] Sonja Merten,[1,2] Kaspar Wyss[1,2]

[1]Swiss Tropical and Public Health Institute, Basel, Switzerland
[2]University of Basel, Basel, Switzerland

**Correspondence to**
Dr Pauline Yongeun Grimm;
pauline.grimm@unibas.ch

## ABSTRACT

**Objective** The aim of this study is to improve the understanding of the characteristics of health system resilience in Myanmar's response to Cyclone Nargis and to explore ways to improve resilience at the system level.

**Design and setting** This is an explanatory qualitative study exploring the institutional capacity of resilience in Myanmar's health system. Analysis proceeded using a data-driven thematic analysis closely following the framework method. This process enabled comparisons and contrasts of key emergent themes between the participants, which later generated key results describing the foundational assets, barriers and opportunities for achieving resilience in Myanmar.

**Participants** The study comprised of 12 in-depth interviews conducted with representatives from international organisations and non-governmental organisations (NGOs). The inclusion criteria to recruiting the participants were that they had directly been a part of the Cyclone Nargis response at the time. There was a balanced distribution of participants across UN, bilateral and NGOs, and most of them were either Myanmar citizens or expatriates with extensive working experience based in Myanmar.

**Results** Key findings elucidate the characteristics of resilience that have been salient or absent in Myanmar's response to Cyclone Nargis. Strong social capital and motivation propelled by its deep-rooted culture and religion served as Myanmar's greatest assets that filled major gaps in the system. Meanwhile, its postcolonial and military legacy posed barriers towards investing in building its long-term foundations towards resilience.

**Conclusions** This study revealed that resilience in the health system can be built through strategic investments towards building the foundations of resilience to better prepare for future shocks. In the case of Myanmar, social capital and motivation, which surfaced as its foundational assets, can be channelled into opportunities that can help achieve its long-term health goals, accelerating its journey towards resilience in the health system.

## INTRODUCTION
### Health system resilience
The 2014–2015 Ebola epidemic in West Africa served as a turning point in global health. Resilience was introduced as a concept relevant in the discourses for health system strengthening as well as for crisis

> **STRENGTHS AND LIMITATIONS OF THIS STUDY**
> ⇒ This study tests the applicability of a conceptual framework for the first time in a lower middle-income country and builds on the health system resilience conceptual framework developed from a previous systematic review.
> ⇒ The topic of health system resilience has a potential to make a timely contribution to the health systems and policy research community, with lessons generated that can be transferred to other contexts.
> ⇒ The study only represents the view of international organisations and non-governmental organisations, restricting the scope of study to the perceptions of a single stakeholder group, which limits the generalisability of the findings.

management in fragile settings.[1] Since then, there has been a plethora of literature trying to define, measure and characterise the concept.[2–4] Resilience is once again receiving more attention than ever with the failure to contain the ongoing battle against COVID-19 pandemic.

Health system resilience is defined as the capacity to prepare for and effectively respond to crises while retaining core health system functions when a crisis hits. An important aspect concerns the capability of the system to reorganise itself to meet the evolving needs of the situation.[5]

Several scholars among many have proposed varying conceptual frameworks in order to elucidate the complex nature of health system resilience. In the aftermath of the Ebola crisis, Kruk *et al*[6] responded to the growing demands of multilateral organisations to illustrate five key characteristics of a resilient health system and a proposed resilience index corresponding to these five characteristics to measure resilience. Blanchet *et al*[7] proposed a new model of resilience as an underlying management and governance capacity to absorb, adapt and transform itself in case of a shock. Gilson *et al*[8] and Barasa *et al*[9] introduced the idea of 'everyday resilience',

highlighting the capacities and resources that are required when delivering health services every day, which have to be built over a longer period to address chronic deficiencies in the health system. Building on these ideas, Grimm et al[10] have synthesised empirical studies from low-income and middle-income countries to confirm Kruk et al's five attributes of resilience and to identify five new characteristics that can serve as foundations to be prioritised in resource-constrained settings to unlock system-level resilience. As Grimm et al's[10] five key foundations of resilience are critical inputs to activate Kruk et al's five key attributes of resilience, the paper has employed the five key attributes to develop the interview guide. The analysis applies Grimm et al's[10] framework composed of 10 characteristics of resilience to test its applicability in Myanmar's specific context.

In addition to the relevant frameworks, the role of the communities is critical when discussing resilience in low-income and middle-income countries. A few empirical studies have revealed the vital role community engagement plays in building trust in the health system, which is in turn intricately linked with effective management of crises[11–13].

### Cyclone Nargis and its legacy for Myanmar

A category three Cyclone Nargis struck Myanmar on the second of May 2008 affecting more than 50 townships and devastating mainly the Ayeyarwady Division and its capital, Yangon. It is recorded as Myanmar's worst natural disasters in recorded history.[14] Within this area, the official death toll was estimated 138 000,[15] but the damage had severely affected at least 2.4 million lives in the most affected regions. Not only did the cyclone result in a widespread destruction of homes and major infrastructure, but close to 75% of the health facilities were damaged in the affected townships. Most of the destroyed facilities were that of primary healthcare, which exacerbated healthcare access to the rural poor and amplified economic inequalities.[14 16] The Myanmar government at the time missed the critical period following the cyclone to implement essential measures to save lives and provide relief efforts.[17] The agreement to issue visas for foreign humanitarian aid workers and allow international assistance came 3 weeks after the storm, only after the UN Secretary-General Ban Ki-moon visited Yangon to plead with General Than Shwe, the head of the military state.[18] To this day, the response to Cyclone Nargis reverberates lessons for the international community.

### Objectives

The aim of this study is to understand how the characteristics of health system resilience have played out in Myanmar's response to Cyclone Nargis and to explore ways to improve resilience at the system level. Using a refined conceptual framework for health system resilience developed previously, this paper analyses Myanmar's assets and barriers to achieving health system resilience and highlights the lessons learnt from its devastating past with the hope that the lessons may be transferrable to other countries.[10]

## METHODS

### Study setting

This is an explanatory qualitative study exploring the organisational capacity of resilience in Myanmar's health system. Myanmar was recently ranked second on the Global Climate Risk Index of countries most affected by extreme weather events from 1999 to 2018,[19] and 17th out of 191 countries on the Index for Risk Management, which assesses the risk of humanitarian crisis and disasters that could overwhelm national capacity to respond.[20] It is prone to various natural hazards such as earthquakes, floods, cyclones, droughts and fires, which have intensified with the effects of climate change.[21]

Myanmar's national health plan (2017–2021) highlights Universal Health Coverage (UHC) as its overarching goal, and the Ministry of Health and Sports has initiated steps to improve compliance to the International Health Regulations (IHR) in the recent years.[22 23] Initiated by the Union Minister of Health and Sports, Myanmar was the third country in the Southeast Asia region to take part in the Joint External Evaluation (JEE) in 2017 to assess the country's capacity under the IHR to prevent, detect and respond to public health threats.[24] In order to follow-up on the recommendations from the JEE findings, Myanmar's Ministry of Health and Sports has developed a 5-year National Action Plan for Health Security.[25] These efforts demonstrate the country's political will to embark on a journey towards building the foundations of health system resilience.

### Study design and participants

The study comprised of 12 in-depth interviews conducted between January and February 2020, with its participants purposively selected through a snowball sampling of relevant referrals via contacts introduced through the author PYG's existing network while working and living in Myanmar.[26] This proved as an effective method to track participants that had been directly involved in the response efforts of Cyclone Nargis in 2008. In order to partially offset a possible recall bias due to the time lag between the actual event and data collection, triangulation through supplementary data such as event pictures, reports and media clips were provided by the participants when available to add to the validity of the data collected. Among the many stakeholder groups, only those who had represented or currently represents international organisations and non-governmental organisations (NGOs) have been included in the interviews. There was a balanced distribution of participants across affiliations and nationalities. There were five UN staff, three bilateral representatives, three NGO workers, six Myanmar citizens and six expatriates. As the interviews targeted a single stakeholder group, the 12 interviews were shown sufficient to reach data saturation.[27]

## Data collection

The interviews lasted between 1 and 1.5 hours, respectively, and were conducted in-person by the author PYG (PhD candidate) in English. There were no prior exchanges between the participants and the data collector, and the interviews were held in private office spaces of the participants. A semistructured interview guide was developed to explore the roles of the participant, actions taken as an organisation and perceptions of the strengths and weaknesses in the health system at the time of the crisis. The questions were framed around Kruk *et al*[5]'s five dimensions of resilience. See online supplemental file 1 for the full interview guide. All interviews have been audiorecorded following informed consent and thereafter transcribed verbatim. Interviewees were approached a second time via video conference in order to verify the accuracy of the transcripts.

## Data analysis

Analysis proceeded using a data-driven thematic analysis, closely following the steps of the framework method.[28] Verbatim transcriptions and field notes were reviewed for familiarisation and verified for accuracy. Following the development of an initial codebook inspired by the 10 themes previously developed by the author's team, all transcripts were coded line by line using MAXQDA 2018. A hybrid of deductive and inductive approaches provided for a broader structure of the categories and flexibility of codes from the open coding process. As illustrated in figure 1, Grimm *et al*[10]'s conceptual framework was used to develop the analytical framework. The framework was then applied systematically to chart the relevant summary of the transcript with direct quotations structured around 10 categories and 21 codes, resulting in a complete framework matrix. This process enabled comparisons and contrasts of key emergent themes between the participants, which later generated key results describing the foundational assets, barriers and opportunities for achieving resilience in Myanmar. A few of the interviewees provided feedback on the final analysis through a subsequent virtual meeting arranged by the author.

## Patient and public involvement

The research was done without public involvement.

## RESULTS

The results section presents key findings elucidating the characteristics of resilience that have been salient in Myanmar's response to Cyclone Nargis. The results are categorised into assets, barriers and future opportunities for achieving health system resilience.

### Foundational assets that can unlock Myanmar's health system resilience

#### Strong social capital comprised informal channels and trusted networks

Most participants pointed to the extensive role that the informal sector plays in filling the gaps of the health system. A case in point, the ambulance system in Myanmar is currently charity based and subjected to resource availability from various donors. The township hospitals themselves have to seek out appropriate charity groups to maintain the referral system composed of ambulances

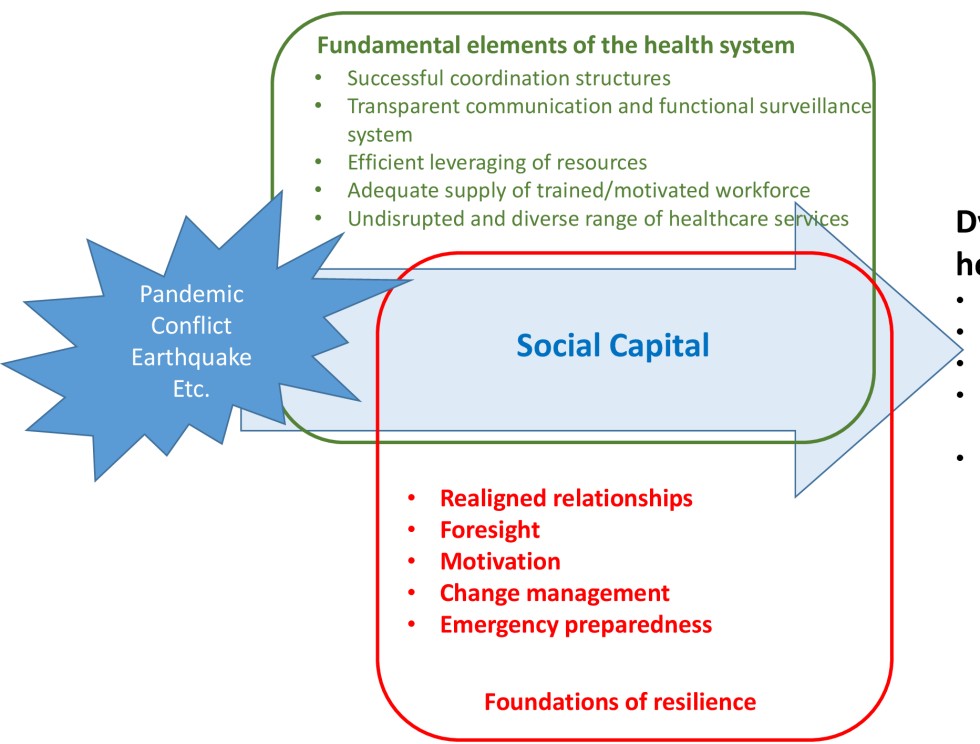

**Figure 1** Grimm *et al*'s[10] health system resilience framework.

and drivers. Likewise, when Cyclone Nargis hit, there were various community-based organisations as first responders of the crisis, offering food, shelter and basic commodities to the affected villages. Strong social capital in Myanmar were exhibited mainly through informal channels, whether it be via monasteries and community-based charity groups. They were able to operate without legal mandates and special memorandum of understanding, precisely because they have obtained people's trust and leveraged the familiarity of their own network. One NGO worker described:

> There was an existing social community that was ready to donate to religious events and they were actively involved and participated in the relief efforts after Nargis hit. Each village has a social committee and the religious leaders in the monastry have their own shelters for the affected people who sought refuge. The communities themselves know their needs and gaps the best, which can build resilience. (I09)

In a context where access to the affected regions were limited for the first 3 weeks after the cyclone, community engagement proved vital in offering timely support. It was apparent that participants who visited the affected areas experienced severe logistical challenges. Some of the areas were only accessible by boat or in off-road conditions where a robust 4×4 vehicle was required. In this endemically challenging environment, mobile clinics were established to reach the communities in remote areas. All the more in these contexts, community-based resilience was not optional but imperative.

> People had to travel long distances to access health care, which severely limited preventative care and people accessing the health system…One thing that consistently stood out to me was the incredible self-sufficiency of the communities themselves, their resilience, and their ability to get back on their feet with the limited resources that they had. (I11)

Referring back to the example of charity-based ambulances, a few participants stated their concerns that this heavy reliance on informal channels can in turn be a barrier towards building basic systematic preparedness, as often these groups can run without standardised procedures and beget further inequalities in service delivery.

> A basic preparedness however is not equipped because of this reliance on informal channels. Not all township hospitals have ambulances. Apparently, there is no standard. An ambulance and a driver are not part of a township health department set up. (I06)

As evidenced through Cyclone Nargis, social capital although informal in nature served as an asset for Myanmar in both crisis and calm.

## Cultural and religious commitment as a propeller for motivation

Underpinning the strong social capital lay the commitment of community-based groups as the backbone of relief efforts. Many attested that Myanmar's history of prolonged isolation and underdevelopment led to a stronger demonstration of resilience at the community level especially in a situation of emergency. In Myanmar, volunteerism is a common method of civic participation.[29] The motivation for this commitment appeared to have stemmed from the people's religious and cultural beliefs to give to charity and donate to the less fortunate, which also manifested in the case of Nargis. A UN staff member shared:

> People in a community supporting other people in need is a powerful response. Most believe that it's because they were closed off for so long, and it was just helping themselves, and the nature of the general beliefs and teachings of the country. People will go to churches, pagodas, temples, monastries or other public buildings to shelter, and the community there will provide food for a couple of days generally. (I07)

Traditional social welfare support groups revolving often through informal channels played a vital role at the village level, and many self-help groups sprung up spontaneously to join the momentum.[30] The monasteries even had an informal system to store food collected through donations and shared them with the communities in need. They were the sole sources of aid in the immediate aftermath of Cyclone Nargis until support arrived from the government and UN agencies, which proved as a clear asset. Several interviewers echoed that these social determinants of religion and culture provided a supporting environment for subsequent relief projects to flourish.

> The Buddhist culture compels people to donate to those in need and the monks. The first responses have always come from the people/monasteries, later from the government and finally from the UN agencies. It's because it's part of the culture, all these donations. (I01)

> There are social determinants at play in the response to Nargis. At the time of the disaster, 90% of the people were Buddhist. Those from non-affected townships even came. Came to donate. This is the culture of Buddhism. Supporting environment was religion. (I05)

Community resilience in Myanmar seems to have been built on a stronger foundation because it stems from religion and culture as a motivating force. The powerful source of solidarity within some of the affected communities weathered through generations of political and ethnic divisions.

## Barriers to Myanmar's journey towards resilience

### Historical legacy affecting transparency, openness and decentralisation

When reflecting on the aftermath of the cyclone, all participants called out the lack of transparency and openness in how the government had dealt with the crisis at hand. The government at the time did not convey the exact situation analysis to the public, withheld key information about the cyclone and controlled data such as the degree of damage and casualties incurred. Such lack of transparency undermined public trust towards the government and limited international response teams from accessing Myanmar, which resulted in an overall delay in collective response efforts.

> The government at that time was so quiet and the civil society has not seen the first response from the government. INGOs were not able to access the area until three weeks after the cyclone hit. Perhaps, this is the reason why the government also tried to hide the information as much as possible, for instance, the number of casualties, number of shelters required, etc. We learnt the lack of government information is tied to the lack of accountability. (I10)

Though the reasons may be complex, most respondents pointed to Myanmar's long history of military rule that has operated based on a rigid hierarchy and centralised decision-making system, a context that may not have been conducive for disaster management. There was no proper disaster management mechanism in place; rather, coordination was led through the health cluster. As it had been a highly centralised system, however, the senior general still had the final say and was initially not open to collaborating with the outside world.

> Everything had to be reported and decided by the senior general (Than Swe)… Whatever he said, everybody has to do it. (I04)

To this day, the memorandum of understanding is signed at the national level, not with the state. The requests may be made locally, but the final say comes centrally. One participant stated that this hierarchical structure produces dissonant voices at different levels, complicating the decision-making process and posing a risk of not meeting the actual needs on the ground. Another participant echoed this sentiment:

> Myanmar is too centralized, which poses a key challenge towards building a resilient health system. Evidence-based planning at the local level is not possible. For example, prevention of the polio outbreak that occurred in Hpa-bun would have been possible if local collaboration with the ethnic organisation could be done at the state level, where states have the decision making power and have fiscal decentralisation. (I06)

Many participants pointed to the added layer of complication from the presence of ethnic armed groups in self-administered zones that are run independently of the government as a result of its ongoing strife towards independence.[31] These special subnational administrations run by multiple ethnic groups have their own health system parallel to that of the Myanmar government and selectively collaborates with local NGOs and community-based groups where there are no conflicts of interest. One participant shared that the polio outbreak in May of 2019 in one of these self-administered regions was particularly difficult to contain due to lack of information shared and restricted access to the area.

> Often ethnic health organisations work in these areas to provide services, and some are in fact coordinating well with the ministry. But, access can be difficult, as in the case of the polio outbreak, and the ethnic health organisation wouldn't let the ministry in nor UN agencies. They only allow one NGO they trust to have access. If you're looking at some kind of outbreak, there is a needed preparedness element by the ministry, but they also need to consider this other factor that is not something that other countries should necessarily worry about. (I06)

Often UN agencies and NGOs offer to bridge these gaps in services, yet these two parallel systems pose great challenges for international actors to manoeuvre through the bureaucracies in these conflict zones. It is an ongoing challenge that Myanmar needs to overcome in order to reach an inclusive state of resilience at the national level.

### Lack of investments in long-term variables of resilience

Cyclone Nargis revealed the fragility of its health system, especially the areas that had required long-term investments. One participant that had worked with the UN agency during this time depicted how dire the situation had been.

> The basic fabrics of the health system were not existing. There was not a place to even fit supplies, medicines and computers. Many RHCs were not functioning well, many times lacking midwives and essential medicine. The nurse to patient ratio was too low and building the nursing infrastructure was critical. (I04)

According to these first-hand accounts, Myanmar still had a very basic infrastructure, township hospitals covering up to 200 000 people and a rural health centre serving around 20 000–25 000 people.[14] A typical rural health centre had a midwife, but this was not necessarily the case in remote and hard-to-reach villages. Health worker shortages came up in all the interviews as a major obstacle for Myanmar's achieving its Universal Health Coverage goal, let alone its journey towards resilience. One UN staff indicated that on average only 54% of the sanctioned health worker posts are filled in Myanmar, and building a strong human resource cadre that is evenly distributed across the health system requires a conscious long-term commitment and investment from the government.

Though development cooperation can support buildings, supplies and equipments, robust human resources require a long-term commitment and investment that should come from the government…I mean, the largest issue of Myanmar is that you have only 54% of sanctioned posts filled. Even the 54%, if you see differences in regions. (I06)

All participants stressed the importance of investing in long-term infrastructure, including its health workforce across the country, particularly focusing on extending its health system coverage to remote and vulnerable areas in order to build resilience of the health system as a whole. However, the stark inequalities in remote areas and the frequent staff turnover at all levels of the government were reflections of poor working conditions and weak human resource policies. Participants currently working for a major UN agency shared that funding additional trainings may not be the solution to this health workforce challenge.

Because of the weakness in the human resource policy, and the working nature of the government job, many government staff are resigning, quitting their jobs. We are losing qualified candidates. Human resource is a major challenge…I know many trainings were supported by organisations and donors, but the problem is the retention of human resource. After the trainings, they move or resign. I think training should not be the key weakness. (I02)

Some specifically referred to the importance of foresight for these long-term commitments. There seems to be a significant gap between plans on paper and implementation as mentioned by one civil society representative:

The challenge in the state/region is human resource shortages, especially medical doctors. The analysis of the ministry of health national health plan reveals that across the country there is a health service gap due to the lack of human resources. There are not enough medical doctors trained through the medical universities. The government already has a plan, but there is a big implementation gap. (I10)

### Myanmar's future for resilience
#### Harnessing social capital to unleash the power of collective civic duty through civil society groups
As echoed by multiple participants, Myanmar appears to have the ingredients for resilience within its own respective communities. Not only did existing community-based groups act as first responders of Cyclone Nargis although decades of dictatorship, but even the coordination took place naturally and provided mutual support. It was simply the people exhibiting civic duty and playing their due role in the society. One representative of a local civil society group shared that health system resilience may be reached when civil society groups build community resilience and start working together with a health system that is also investing in building its strengths.

At the time of the cyclone, the local civil society groups in each area organised and coordinated with each other by themselves. We haven't seen any government, like red cross, or police. We haven't seen any solider. During the Cyclone Nargis, we have seen the communities, the people, how did they respond. They came out to the street and they supported each other. They tried to remove all the trees and blocks. We learnt from them. We have strong resilient communities. Communities know best what the biggest challenge and needs are in those areas. If we work together, community resilience strengthening and health system strengthening, we could be working towards resilience. (I10)

The informal channels of self-formed community-based groups can be a strong basis for the institutionalisation of civil society groups that have a rather complicated relationship with the government. Since Myanmar has opened up in 2011 under President Thein Sein, its reform process has gradually enabled a platform for civil society to establish civil liberties through its '2008 Constitution', with the article 354 stating the right to freedom of expression, assembly and association.[32] Furthermore, the Parliament has enacted the Association Registration Law, which defines and clarifies local and international non-governmental organisations and stipulates requirements to register with the government. These reforms have gradually shifted the landscape of civil society engagements at all levels of the country.

Even at the national level, the government has been more receptive to the inputs of the civil society groups, including the Ministry of Health and Sports with many of its community-driven development projects. There are certainly growing opportunities for civil society groups to participate in the various development sector working groups.[33] One participant reflects on the evolution of civil society engagement since Cyclone Nargis and projects that no crisis can be handled without the involvement of the country's civil society groups.

The health system at that time was already weak and the ministry of health didn't have a strong coordination with the civil society or community groups. The government organised many meetings but had no idea how to work with the local civil society. Only in the last four years did this relationship change. The minister of health stressed the importance of working with CSOs across the country. If an outbreak were to happen, the ministry of health cannot handle the situation without civil societies. (I10)

### Connecting its overarching health goals with long-term investments to improve resilience
As Myanmar is situated in a disaster-prone area, it has naturally developed a local expertise over time through handling its yearly floods. One participant stated that there is an emergency fund stipulated by law to set aside

a disaster management fund that can be used to support subnational levels in case of a potential crisis. How the legal frameworks and plans align with its budgeting and allocation, however, is a different issue.

A recurring theme that transpired when discussing the government's response to the cyclone and other subsequent crises were the emergence of discrepancies between planning, budgeting and implementation. As there has been no clear alignment between its long-term health policy and crisis management, the targeted investment towards building system-level resilience has also lagged behind.

> There is very remote connection between planning and budgeting. It happens both at the national and local levels. Preparedness is there on paper. (I06)

One participant called this the ability to 'expand and contract', which also corresponds to the concept of adaptiveness in Kruk *et al*'s framework. It is exhibited by the ability to reallocate its resources to where it is most needed on time and to delegate the authority to the appropriate decentralised administration when necessary.[6] It is a vital characteristic of resilience, which is a byproduct of long-term investments and painstaking reforms.[34]

> This kind of planning, the constant expansion and contraction that you would need in an environment like Myanmar that's so natural disaster heavy, is difficult. (I07)

Many participants agreed that Myanmar ought to invest towards building long-term infrastructure especially in remote areas and prioritise in its human resource preparedness in order to build resilience in the system for future shocks. This may involve employing creative ways to increase staff retention through the engagement of the private and non-profit sectors. More importantly,

however, these plans ought to connect back to its overarching national goal for Universal Health Coverage and its momentum to strengthen its commitment towards International Health Regulations.

> In order to improve health system resilience, it is necessary to improve infrastructure needed at the village level… A challenge in building the long-term capacity of health workers, however, is the frequent turnover of staff. Mostly staff assigned for the rural health centres are based there for 1–2 years and move to a better place. Turnover was quite quick. (I09)

## DISCUSSION

This study explored the degree to which key characteristics of health system resilience manifested in Myanmar's response to Cyclone Nargis and their respective historical and sociopolitical determinants. As health systems are complex and evolving in nature, it is important to predicate this study on the view that resilience is a capacity that can be nurtured with time and commitment.[35] Moreover, facets of resilience should not be seen as disconnected inputs, but rather a holistic result of targeted investments and a series of structural reforms.[10 34]

In the case of Myanmar, it had been the 'everyday resilience' of its frontline workers and communities that had been built through its day-to-day chronic struggles in the context of its weak governance and political and economic vulnerabilities that in turn prepared them for the unexpected shock.[8] As depicted in figure 2, key themes that emerged from the analysis have been organised into assets, barriers and future opportunities in describing Myanmar's journey towards resilience.

First, social capital and motivation have surfaced as key foundational assets in Myanmar that are indicative of its

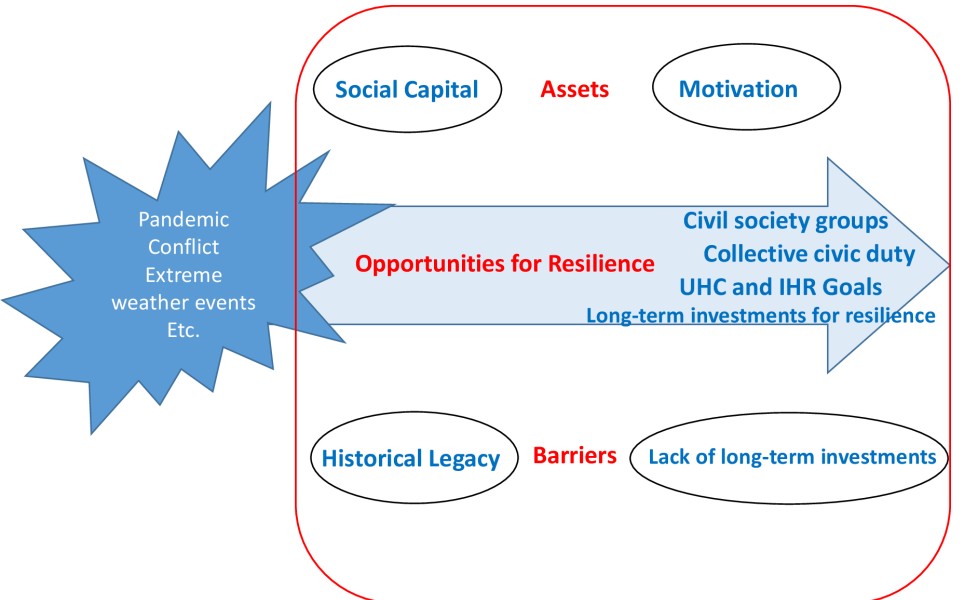

**Figure 2** Assets, barriers and opportunities for Myanmar's journey towards health system resilience.

community resilience. The power of collective civic duty filled the service gaps that the government had failed to provide in the immediate aftermath of the cyclone. The social capital present in Myanmar during Cyclone Nargis were composed of informal channels and trusted networks in the communities, its motivation stemming from a deeper faith rooted in religion and culture. A recent qualitative synthesis highlighted social capital as the interconnecting mechanism common to both health system strengthening and resilient responses to crises, a powerful tool that can unlock resilience of the health system.[10] This study extends the notion a step further by demonstrating that the power of social capital can go beyond its health system capacity. Social capital may in fact play a more important role in providing relief from the cyclone, just as it had been reported as being more vital in mitigating the spread of the Ebola virus than any donor-driven short-term interventions.[13] The pivotal role social capital plays in building community resilience have been well documented by other studies in the context of both natural and man-made disasters.[36–40] When communities themselves start carrying out critical health system functions, this collective momentum for community resilience prompts what Barker *et al*[11] calls a 'fortuitous cycle' of increased trust, improved communication and continued meaningful community engagement, all of which are crucial preconditions for enabling health system resilience.

Second, Myanmar's historical legacy of prolonged military rule and a lack of long-term investments posed as major barriers to achieving resilience. The lack of transparency and openness stood out as the biggest obstacle towards crisis management, which undermined public trust towards the government and hampered international response teams from accessing Myanmar in the initial stages. Myanmar is a prime example of a post-colonial state that development experts describes as a 'conflict trap'.[41] The legacy of the colonial experience, ethnic insurgencies and a successive military rule shifted all of its government's priorities to regaining its national sovereignty and ensuring security through increased military spending.[42] Consequently, little had been left for health and social welfare, with only approximately 2% of its Gross Domestic Product (GDP) spent on healthcare shortly before the cyclone hit.[30] It is no surprise that Myanmar's health system was fragile and not ready to handle a disaster of such magnitude.

Governance practices influence organisational resilience and decentralised governments are certainly at an advantage to exhibit the necessary flexibility and timely responses required in crisis management.[2] Having a decentralised governance structure itself, however, does not guarantee a successful response. China had no national disaster management agency nor related policies during the Sichuan earthquake but effectively controlled the emergency with its president's charismatic leadership and vertical administrative structure.[43] Myanmar's centralised and hierarchical system could have

been used for Myanmar's advantage, but an adequate disaster management had not been the priority of the government.

A common pattern across failed attempts in responding to a crisis was to prioritise on the 'fast variables', delivering services in the form of temporary shelters and supplies without considering its longer term infrastructure building, as it had been in the case of Liberia during its Ebola crisis.[44] Building resilience, however, is much more than preparedness, as Kruk *et al*[6] states, that it requires investing in systems and its corresponding 'slow variables' that can function in both crisis and calm. A health system that has a clear foresight for its people will strengthen its health system infrastructure and train its human resource cadre to build the foundations of its health system.[10] In the context of Myanmar, it would mean to apply a systems thinking approach in examining its past shortcomings and redesigning its institutions and organisational structures to serve its evolving purpose and prepare for future shocks.[45]

Third, Myanmar may channel its assets into opportunities by using its growing civil society platforms and leveraging its long-term national health goals. Cyclone Nargis revealed that Myanmar has the key ingredients for resilience within its own communities through its people exhibiting their civic duty. Civil society has had a long history in Myanmar dating back to its informal Buddhist associations in the traditional kingdoms and various other religious groups organising social and welfare programmes during its colonial period and beyond.[46] Since Cyclone Nargis, the space for civil societies further expanded through its legal reforms, but at the end of the day, only a few have been able to obtain official registration and secure a sustainable funding mechanism.[46] According to the Civil Society Survey conducted in 2019, despite its marked progress, a wide range of obstacles still remain and hinder civil societies from reaching their goals.[32] Myanmar has set ambitious national goals to reach UHC and to improve compliance to the IHR in order to ensure healthcare and health security for all.[22 25] As evident in its neighbouring countries, however, civil society engagement and community feedback loops for policy and decision-making processes have shown as prerequisites to accelerate progress towards achieving these compendium of goals.[47] Trust and accountability at all levels in the system are what builds the 'capacity of legitimacy', a crucial component of resilient health systems, that can only be strengthened through an inclusive consultation process of engaging communities and civil society groups.[7] Myanmar has to realise that strategically leveraging these opportunities may be the fastest route to building resilience.

The findings of this study offer tangible suggestions for Myanmar to invest on building its hard and soft long-term infrastructures in remote areas, prioritising in human resource preparedness and redesigning institutions and organisational structures to serve its evolving purposes and to prepare for future shocks.

## Study strengths and limitations

This study is empirical in nature, building on a health system resilience conceptual framework developed from a previous systematic review. The framework has been tested for the first time for its applicability in a lower middle-income country. Regarding the study design, the long time lag between the actual event and data collection introduces a potential recall bias, which was partially offset by triangulation through supplementary data such as reports, event pictures and media clips provided by the participants. The topic of health system resilience has a potential to make a timely contribution to the health systems and policy research community, with lessons generated that can be transferrable to other contexts. Due to COVID-19 pandemic's travel restrictions imposed during the data collection period, the study only represents the view of international organisations and non-governmental organisations, restricting the scope of the study to the perceptions of a single stakeholder group. Further studies should explore capturing a wider range of stakeholders, the perceptions of communities, frontline workers and government staff that may be able to offer a more holistic view of the phenomenon.

## CONCLUSION

This study revealed that resilience in the health system can be built through targeted investments towards building the foundations of resilience to better prepare for future shocks. In the case of Myanmar, social capital and motivation, which surfaced as its foundational assets, may be channelled into opportunities that can help achieve its long-term health goals, accelerating its journey towards resilience in the health system.

**Acknowledgements** The authors would like to extend appreciations to all the interviewees who carved out time to part take in this study. Special thanks to Dr Nilar Tin for her valuable input on the initial study concept and for suggesting relevant interviewees.

**Contributors** PYG and KW conceptualised the study; PYG collected and analysed the data with substantive input from SM and KW. PYG developed the first draft of the manuscript, and all authors contributed to the subsequent and final drafts.

**Funding** This article is part of the PhD project that has received funding from the European Union's Horizon 2020 research and innovation programme under the Marie Skłodowska-Curie grant agreement No. 801076, through the SSPH+ *Global PhD Fellowship Programme in Public Health Sciences (GlobalP3HS) of the Swiss School of Public Health*. The corresponding author is also a recipient of the Swiss Government Excellence Scholarship (ESKAS), provided by the Swiss Federal Commission for Scholarships for Foreign Students (FCS) with the award number 2018.0217.

**Disclaimer** The funding bodies did not play a role in the design, analysis, and writing of the manuscript.

**Competing interests** None declared.

**Patient consent for publication** Not required.

**Ethics approval** The proposal for this study was approved in Myanmar by the Institutional Review Board of the University of Public Health and the Ministry of Health and Sports of the Republic of the Union of Myanmar (2020/Research/4). It has also received clearance in Switzerland by the Ethikkommission Nord-west und Zentralschweiz (Req-2019-00348).

**Provenance and peer review** Not commissioned; externally peer reviewed.

**Data availability statement** All data relevant to the study are included in the article or uploaded as supplementary information. This is a qualitative study of a small number of interviewees representing international organizations and NGOs. All data relevant to the study are included in the article or as supplementary information. Making the full data set publicly available could potentially breach the privacy commitment made to the participants upon obtaining informed consent, as well as the ethics approval granted by both the Institutional Review Board of the University of Public Health in Myanmar and Ethikkommission Nord-west und Zentralschweiz in Switzerland. Therefore, the authors will not make the full transcripts available to a wider audience.

**ORCID iD**
Pauline Yongeun Grimm http://orcid.org/0000-0002-8101-9197

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
