## [Reviewer comments · BMJ Open]

ARTICLE DETAILS

TITLE (PROVISIONAL)	Evidence of health system resilience in Myanmar during Cyclone Nargis: a qualitative analysis
AUTHORS	Grimm, Pauline Yongeun; Merten, Sonja; Wyss, Kaspar

VERSION 1 – REVIEW

REVIEWER	Thidar Pyone Public Health England, Department of Global Health
REVIEW RETURNED	17-May-2021

GENERAL COMMENTS	General comments  [ ] A very interesting and well-written paper, using a conceptual framework to explore experiences in response to a natural disaster (cyclone Nargis) in Myanmar [ ] The paper uses the qualitative research method to explore opinions from international stakeholders who were reported to be part of the response [ ] The study missed important stakeholders' group of the health system (government officials, policymakers and frontline health workers). Nevertheless, the authors have acknowledged that in the limitation section. [ ] The study used Kruk's et al's framework to guide the development of interview questions and used a different framework (Grimm et al.) for analysis. Why there are two frameworks, and how they interlink? [ ] The study also produced its own framework based on its findings, which is illustrative and useful. [ ] It is very timely to study health system resilience, especially during the long-standing pandemic. However, one key concern is "recall bias" as Cyclone Nargis happened in 2008, and the study explored experiences and opinions of people 12 years aftermath. There is a long-time lapse between the actual event and the data collection period. Besides, the authors did not mention anything about which measures they took to control those recall bias? [ ] The authors provide an interesting account on the response, linking to domains in health system resilience. However, there's little information on how these findings will be helpful for preparedness in future shocks. The reviewer noticed a specific section called "preparedness and future shocks" in the interview guide. But the findings and discussions focused only on response. Discussion on this aspect will be helpful for a disaster-prone country like Myanmar. Specific comments  1. Abstract  [ ] Line 41, Page-1-"low-income country" should be changed to "lower-middle-income country". 2. Introduction section  [ ] The sentence on Line 77-78 of page 2 is difficult to understand. Suggest rephrasing.
--

	[ ] Which region does the author refer? Line 86 of page-2—affected regions (Ayeyarwady and Yangon)? Need to clarify [ ] The term “Burmese” on Line 89 of page-2 should be replaced with “Myanmar”. It should be “The Myanmar government”, not “the Burmese government”. 3. Results section Line 280 of Page-7 stated “recent polio outbreak...”. Recent means recent to the data collection, which was between 2019-early 2020? Not recent to Cyclone Nargis’ incidence time which was 2008?
--	---

REVIEWER	Kyu Kyu Than Burnet Institute
REVIEW RETURNED	19-May-2021

GENERAL COMMENTS	I would like to congratulate the authors for a comprehensive analysis using the health system resilience framework based on the real crisis situation in Myanmar (Cyclone Nargis). My comments are as follows: 1. Although the findings and analysis are very relevant to the Myanmar context in general, I have doubt in the participants response after 12 years of Cyclone Nargis (2008). They may have brought some of their views on the current health system rather than the time of Nargis. My suggestion is that this should be describe well in the weakness of the study section. 2. In line 384: I am not clear on what you mean by "Preparedness is on paper and so far there were no big loss in human lives" Please kindly clarify. 3. In line 275-278 "An added layer of complication is the presence of ethnic armed groups in self-administered zones that are run independently of the government as a result of its ongoing strife towards independence" seems more like an authors voice rather than the participants voice. The issue of ethnic armed organization does not seem to fit in with the topic. I wonder that participants are stating the general impression of the overall health system fragility and the system challenges. 4. I find the discussion section very interesting with good analysis towards health system resilience and its future perspectives for the Myanmar health system.
--

REVIEWER	Janet Long Australian Institute of Health Innovation, Australian Institute of Health Innovation
REVIEW RETURNED	22-Jun-2021

GENERAL COMMENTS	Thank you for the invitation to review on the timely and important topic of health system resilience. I found this a really interesting paper examining Myanmar's response to the Cyclone Nargis disaster in 2008 from a health system resilience framework and exploring ways they could improve resilience. The rationale for the study was compelling - not the least of which is Myanmar's status as one of the highest risk countries in the world for climate-related disasters. Twelve in-depth interviews conducted between January and February 2020 were undertaken with representatives from NGOs, UN agencies etc. Unfortunately, planned interviews with government officials and local services was not possible due to COVID restrictions. Overall the paper was well written with a clear structure and robust methodology.
---

	Methods: The interviews were conducted with rigour. The schedule was developed using Kruk’s framework and were well suited to the topic under examination. Results showed a sound analysis and key points were illustrated with appropriate quotes. Discussion shows a deep understanding of disasters elsewhere and of Myanmar policies and priorities that contribute to limit resilience. It is rare that I have no suggestions for authors I review. Thanks for this excellent work. I look forward to seeing more on this very important topic in health services research.
--	--

VERSION 1 – AUTHOR RESPONSE

Reviewer 1 – Comments	Our Response
Dr. Thidar Pyone, Public Health England	
General Comments The study used Kruk’s et al’s framework to guide the development of interview questions and used a different framework (Grimm et al.) for analysis. Why there are two frameworks, and how they interlink?	Thank you for this excellent point. The two frameworks are intimately linked in a way that the five key foundations of resilience (Grimm et al) serve as vital ‘inputs’ for the five key attributes of resilience (Kruk et al). Therefore, we have employed the five key attributes of resilience (Kruk et al) for the questionnaire in an effort to take an unbiased approach to what might have been the key ingredients behind resilience in Myanmar’s specific case. The analysis stage, however, applied Grimm et al’s framework after the data has been collected, in order to test this new framework. So to better clarify this we made this relationship more explicit and added in the background section an additional clarification. Please see pg. 2 lines 76-79.
It is very timely to study health system resilience, especially during the long-standing pandemic. However, one key concern is “recall bias” as Cyclone Nargis happened in 2008, and the study explored experiences and opinions of people 12 years aftermath. There is a long-time lapse between the actual event and the data collection period. Besides, the authors did not mention anything about which measures they took to control those recall bias?	This is indeed a crucial point which has not been adequately described in the manuscript. We have inserted therefore under the methods (study design/participants section) clarification on the measures we have taken to partially offset recall bias through triangulation. In addition, we have also added recall bias as a limitation of the study under the study strengths and limitations section. Please find these additions on pg. 3 lines 133-136 and on pg. 12 lines 504-507.
The authors provide an interesting account on the response, linking to domains in health system resilience. However, there’s little information on how these findings will be helpful for preparedness in future shocks. The reviewer noticed a specific section called “preparedness	Thank you for this insightful comment. As the reviewer correctly points out, there is a “preparedness and future shock” section in the interview guide and the responses have been embedded throughout the results section under “Myanmar’s future for resilience” and the

and future shocks” in the interview guide. But the findings and discussions focused only on response. Discussion on this aspect will be helpful for a disaster-prone country like Myanmar.	discussion section. For example, as a result of the analysis, this paper provides selected tangible suggestions (e.g. to invest on building long-term infrastructures in remote areas and prioritize in human resource preparedness, and furthermore to redesign institutions and organizational structures to serve its evolving purpose and prepare for future shocks.) To highlight and summarise these aspects and tie them back to practical steps for preparedness against future shocks, we have added an extra paragraph in the discussion section on pg. 11, lines 497-500.
Specific Comments Abstract Line 41, Page-1-“low-income country” should be changed to “lower-middle-income country”.	Thank you for this correction. We have changed on both occasions the terms to “lower-middle-income country” on pg. 1, line 42 and on pg. 12, line 504.
Introduction The sentence on Line 77-78 of page 2 is difficult to understand. Suggest rephrasing. Which region does the author refer? Line 86 of page-2—affected regions (Ayeyarwady and Yangon)? Need to clarify The term “Burmese” on Line 89 of page-2 should be replaced with “Myanmar”. It should be “The Myanmar government”, not “the Burmese government”. Results Line 280 of Page-7 stated “recent polio outbreak...”. Recent means recent to the data collection, which was between 2019-early 2020? Not recent to Cyclone Nargis’ incidence time which was 2008?	The sentence has been rephrased on line 81-82 of pg. 2 to improve clarity. We have added “most affected regions” to clarify that we are referring to these two regions. Please find this on line 91 of pg. 3. “Burmese” has been replaced with “Myanmar” on line 95 of pg. 3. “recent polio outbreak” has been replaced with “the polio outbreak in May of 2019” on line 289 of pg. 7.
Reviewer 2 – Comments Dr. Kyu Kyu Than, Burnet Institute	Our Response
General Comments Although the findings and analysis are very relevant to the Myanmar context in general, I	Thank you for this crucial point also raised by reviewer 1 which have not been adequately elaborated in the manuscript. We have inserted under the methods (study design/participants

have doubt in the participants response after 12 years of Cyclone Nargis (2008). They may have brought some of their views on the current health system rather than the time of Nargis. My suggestion is that this should be describe well in the weakness of the study section.	section) details on how we have partially offset the possible recall bias through triangulation of multiple sources. In addition, we have included recall bias as a limitation of the study under the study strengths and limitations section. Please find these additions on pg. 3 lines 133-136 and on pg. 12 lines 504-507.
Specific Comments In line 384: I am not clear on what you mean by "Preparedness is on paper and so far there were no big loss in human lives" Please kindly clarify.	This is certainly a valid point and this phrase has been directly taken as a verbatim quotation from one of the participants, albeit its lack of clarity. The key message was that according to the participant's experience/viewpoint, there seems to be a gap between planning, budgeting and implementation. As the part about human lives are irrelevant to this particular topic, we will abridge this quotation to prevent confusion for the readers. Please find this change on pg. 9, line 393.
In line 275-278 "An added layer of complication is the presence of ethnic armed groups in self-administered zones that are run independently of the government as a result of its ongoing strife towards independence" seems more like an authors voice rather than the participants voice. The issue of ethnic armed organization does not seem to fit in with the topic. I wonder that participants are stating the general impression of the overall health system fragility and the system challenges	We apologise that the issue of ethnic armed organisation has caused confusion. Nevertheless, this topic has been a prominent one, fitting under the code "political, ethnic tribalism and armed groups," and repeatedly mentioned by almost half of the participants. As we have reflected upon Nargis as a case study but asked further questions on the participants' experience/viewpoints on what may be a major obstacle for Myanmar's health system resilience (as of now), the presence of parallel structures and health systems in these self-administered zones were seen as a great challenge that Myanmar should consider when discussing resilience building. We have rephrased the sentence on pg. 7 lines 283, however, in order to better convey that this topic has come from the participants themselves and not from the authors.
Reviewer 3 – Comments Dr. Janet Long, Australian Institute of Health Innovation	Our Response
Thank you for the invitation to review on the timely and important topic of health system resilience. I found this a really interesting paper examining Myanmar's response to the Cyclone Nargis disaster in 2008 from a health system resilience framework and exploring ways they could improve resilience. Overall the paper was well written with a clear structure and robust methodology..... It is rare that I have no suggestions for authors I review. Thanks for this excellent work. I look	We would sincerely like to thank the reviewer for taking the time to review and comment on the manuscript. We are very happy to see that the reviewer considers this manuscript well -written and that it would pique the interest of a global audience. Thank you.

forward to seeing more on this very important topic in health services research.	
--	--

VERSION 2 – REVIEW

REVIEWER	Thidar Pyone Public Health England, Department of Global Health
REVIEW RETURNED	28-Aug-2021

GENERAL COMMENTS	Congratulations to the authors on the revision. Some remaining comments to address are listed below. They can also be seen in the manuscript (MS word version). 1) Introduction Page 3, Line 79-80. To avoid confusion to readers who are unaware of Grimm et al.'s framework composed of 10 attributes, suggest rephrasing like "The analysis applies Grimm et al. (2021)'s framework of composed of 10 attributes". 2) Methods: Study setting Page 3, Line 111. What do we mean by institution here? Do you mean the rules of the game as defined by Douglas North (1990) (North DC. Institutional change and economic performance. Cambridge: Cambridge University Press, 1990)? Or is it similar to organisation, the players of the game? If it is organisation, which organisational capacity? Page 4, Line 140. Participants' profile It is unclear what balanced distribution mean? It would be good to describe how many are from the UN and from bilateral organisations and NGOs. Similarly, how many Myanmar citizens and expatriates. Something like this. UN (x numbers), bilateral (x numbers), NGOs (x). Myanmar citizens (x), expats (x). Page 4, Line 162-164. Data analysis Again, for readers who are unaware of Grimm et al.'s framework, it will be useful to have a diagram of the framework so that readers can visualise the analysis framework. Perhaps, the Figure 1 from the discussion can be moved here. 3) Results Page 5. Line 185-189. Strong social capital comprised of informal channels and trusted networks The first sentence, Line 185, is a correct fact but the example (charity ambulance system) given below is not relevant to the time of cyclone Nargis (i.e, 2008). Hence, the example should be replaced as charity ambulances exists only after 2012 when President Thein Sein came into power. His regime had supported the role of CSOs, including charity ambulance systems. This could only happen with increased access to communication means such as mobile phones availability so ambulances can be called upon. In 2008, mobile network coverage was very limited. Page 7. Line 289-292. Historical legacy affecting transparency, openness, and decentralisation The 2019 polio outbreak example in self-administered regions. This is not an example from cyclone Nargis; however, the example is still relevant as the situation hasn't changed. i.e., self-administered regions still have two systems.
---

	4) Discussion Page 10. Line 424-426 and Figure 1 Couldn't find Figure 1 in this document. It's not included in the attached files either. This sentence and figure should move to data analysis under methods.
--	---

VERSION 2 – AUTHOR RESPONSE

Reviewer 1 – Comments	Our Response
Dr. Thidar Pyone, Public Health England	
1) Introduction Page 3, Line 79-80. To avoid confusion to readers who are unaware of Grimm et al.'s framework composed of 10 attributes, suggest rephrasing like "The analysis applies Grimm et al. (2021)'s framework of composed of 10 attributes".	Thank you for this suggestion. We have rephrased this sentence to "The analysis applies Grimm et al. 2021's framework composed of ten characteristics of resilience to test its applicability in Myanmar's specific context." Please find the revision on pg. 2, lines 79-80.
2) Methods: Study setting Page 3, Line 111. What do we mean by institution here? Do you mean the rules of the game as defined by Douglas North (1990) (North DC. Institutional change and economic performance. Cambridge: Cambridge University Press, 1990)? Or is it similar to organisation, the players of the game? If it is organisation, which organisational capacity?	By "institution" we mean "organisation," and specifically the health system's capacity. To avoid confusion, we have changed "institutional" to organisational" on pg. 3, line 109.
Page 4, Line 140. Participants' profile It is unclear what balanced distribution mean? It would be good to describe how many are from the UN and from bilateral organisations and NGOs. Similarly, how many Myanmar citizens and expatriates. Something like this. UN (x numbers), bilateral (x numbers), NGOs (x). Myanmar citizens (x), expats (x).	We have added the participants' specific distribution of affiliations and nationalities on pg. 4, lines 139-141.
Page 4, Line 162-164. Data analysis Again, for readers who are unaware of Grimm et al.'s framework, it will be useful to have a diagram of the framework so that readers can visualise the analysis framework. Perhaps, the Figure 1 from the discussion can be moved here.	
3) Results Page 5. Line 185-189. Strong social capital comprised of informal channels and trusted networks The first sentence, Line 185, is a correct fact but	Thank you for this valid point. As the interview questions were designed to juxtapose both current and past systems, we have included the charity ambulance system as an example of the current referral system still operating with the

the example (charity ambulance system) given below is not relevant to the time of cyclone Nargis (i.e, 2008). Hence, the example should be replaced as charity ambulances exists only after 2012 ...	support of CSOs. The reviewer is absolutely correct that the context was different in 2008, but we hope to keep this example as it connects with the assets that Myanmar has (now) with regards to unleashing resilience attributes in the future. We have added “currently” on pg. 5 line 186 to highlight that this is a recent case example and to take into account the reviewer’s concern.
Page 7. Line 289-292. Historical legacy affecting transparency, openness, and decentralisation The 2019 polio outbreak example in self-administered regions. This is not an example from cyclone Nargis; however, the example is still relevant as the situation hasn’t changed. i.e., self-administered regions still have two systems.	Thank you for pointing this out and acknowledging that we brought in a few recent examples to present Myanmar’s evolving resilience assets and liabilities.
4) Discussion Page 10. Line 424-426 and Figure 1 Couldn’t find Figure 1 in this document. It’s not included in the attached files either. This sentence and figure should move to data analysis under methods.	According to the journal’s guideline, we had included the Figure 1 as a separate file. Please check both Figures 1 and 2 as supplementary files.